

# Using nowcasting technique and data assimilation in a meteorological model to improve very short range hydrological forecasts

Maria Laura Poletti[1], Francesco Silvestro[1], Silvio Davolio[2], Flavio Pignone[1], Nicola Rebora[1]

[1] CIMA Research Foundation
     [2] ISAC-CNR

*Correspondence to*: Maria Laura Poletti (laura.poletti@cimafoundation.org)

**Abstract.** Forecasting flash floods with anticipation of some hours is still a challenge especially in environments made by a collection of small catchments. Hydrometeorological forecasting systems generally allow to predict the possibility
of having very intense rainfall events on quite large areas with good performances even with 12-24 hours of anticipation. However, they are not able to predict exactly rainfall location if we consider portions of territory of 10 to $10^3$ km$^2$ as order of magnitude. The scope of this work is to exploit both observations and modeling sources to improve the discharge prediction in small catchments with time horizon of 2-8 hours.

The models used to achieve the goal are essentially three i) a probabilistic rainfall nowcasting model able to extrapolate
the rainfall evolution from observations; ii) a non hydrostatic high-resolution numerical weather prediction (NWP) model; iii) a distributed hydrological model able to provide a streamflow prediction in each pixel of the studied domain. These tools are used, together with radar observations, in a synergistic way, exploiting the information of each element in order to complement each other: observations are used in a frequently updated data assimilation framework to drive the NWP system, whose output is in turn used to improve the information in input to a nowcasting technique; finally
nowcasting and NWP outputs are blended, generating an ensemble of rainfall scenarios used to feed the hydrological model and produce a prediction in terms of streamflow.

The flood prediction system is applied to three major events occurred on Liguria Region (Italy) first to produce a standard analysis on predefined basin control sections, then using a distributed approach that exploit the capabilities of the employed hydrological model. The results obtained for these three analyzed events show that the use of the present
approach is promising. Even if not in all the cases, the blending technique clearly enhances the prediction capacity of the hydrological nowcasting chain with respect to the use of input coming only from the nowcasting technique; moreover, a worsening of the performance is rarely observed and it is nevertheless ascribable to the critical transition between the nowcasting and the NWP model rainfall field.



## 1 Introduction


Liguria region, located in the north-western Italy, has been recurrently affected in the last century by severe rainfall events that produced flash floods and landslides (Acquaotta et al., 2018). This type of events causes severe damages in the coastal urbanized areas and sometimes also human casualties not only in Liguria Region (Faccini et al., 2009; Faccini et al., 2012; Silvestro et al., 2012a; Davolio et al., 2015; Silvestro et al., 2016) but also in other area of the

Mediterranean (Drobinski, 2014; Delrieu et al., 2004; Ducrocq et al., 2008). All the aforementioned events were associated with well-organized, very intense and localized convective systems affecting the same area of few square kilometres for several hours (Parodi et al., 2012; Rebora et al., 2013; Fiori et al., 2014; Buzzi et al., 2014). A forecast suitable for the small spatial and short temporal scales of these events is needed, but these scales are hardly predictable by meteorological models. Nowcasting models can predict the evolution of the rainfall pattern at regional scale, starting

from the last observed radar rainfall images (i.e. radar-based rainfall nowcasting) and merging different sources of information to provide a short-term forecast (usually a few hours).

Radar-based rainfall nowcasting can be achieved by extrapolating the future rainfall distribution from a sequence of radar images; it has been found that the radar-based rainfall nowcast has good skill for short lead time forecasting. The basic techniques to produce quantitative precipitation forecasts (QPFs) from radar echoes are based on cross-correlation

or individual radar echo-tracking (Collier, 1981). A progress in the nowcasting procedure was the development of nowcasting methods operating in the Fourier domain (Seed, 2003; Xu and Chandrasekar, 2005) to take into account the fact that the predictability of rainfall depends on the spatial scale of the precipitation structures (Wilson et al. 1998; Germann and Zawadzki, 2002). Among the techniques that follows a probabilistic approach, Metta et al. (2009) developed a stochastic spectral-based nowcasting technique that predicts the future rainfall scenarios starting from the

rainfall fields observed by radar end evolving them through a Fourier decomposition. Also, Berenguer et al. (2011) presented a method based on the String of Beads model, while Foresti et al. (2016) analyzed the performance of a nowcasting algorithm that accounts stochastically for the process of growth and decay of rainfall cells.

The main limit of such nowcasting procedures is that the accuracy of rainfall prediction is quite high for very short lead times (20-120 minutes) but, since it is based only on the extrapolation of the observed rainfall field, it rapidly decreases

for longer ranges. As a consequence, also the hydrological forecast is useful in a hydrological nowcasting perspective for limited lead times (Silvestro et al., 2015a) with applications also on urban hydrology (Thorndahl et al., 2017). One of the reason why the accuracy rapidly decreases with increasing lead times is that radar nowcasting techniques do not model (or models them stochastically) processes such as growth and decay of precipitation (Golding, 1998), that become important for longer lead times. So, for very short range (0–3 h), radar nowcasting performs best, whereas for

longer lead times, forecasts based on NWP are better (Kilambi and Zawadzki, 2005; Lin et al., 2005; Kober et al., 2012; Wang et al., 2015). On the other hand, NWP does not allow rainfall predictions with sufficient spatial and time detail (Davolio et al., 2015; Silvestro et al. 2016).

Therefore, after the very first hours (usually 0–3 h) of radar-based nowcasting, NWP forecasts can be merged to generate a seamless 0-6 hour prediction with higher skill. This procedure requires an accurate QPF in the very short

term from the meteorological model, at a high resolution of a few kilometers, since the tolerance for timing or location errors is very limited, especially in case of severe storms (Sun et al., 2014). In fact, a correct forecast allows a smooth transition from radar extrapolation to model prediction. To meet the nowcasting requirements, NWP models have to be





run at convection-permitting resolution (1-4 km, Weisman et al., 2008; Kain et al., 2006) starting from a better initial condition that also reduces the spin-up period. Therefore, several methodologies for rapid data assimilation have been

developed in order to be suitable for nowcasting application. Rapid Update Cycle procedures have been widely used in operational framework (Wilson and Roberts, 2006; Benjamin et al., 2004) to provide a "warm start" and hence reduce the model spin-up. Also, radar reflectivity has been employed to improve the initial condition, e.g. exploiting the information on latent heating through the application of nudging technique (Sokol and Zacharov, 2012; Dow and Macpherson, 2013; Bick et al., 2016; Davolio et al., 2017a), eventually reducing the intensity/position error in rainfall

prediction. Also other studies attempted to improve the hydrological forecast by improving the rainfall prediction using both observations and models. Rossa et al. (2010) assimilated radar data in a NWP system; Davolio et al. (2017a) assimilated rainfall field derived from both radar and gauge observations (Sinclair and Pegram, 2005) in a convection permitting NWP model and used the rainfall prediction in a probabilistic hydrometeorological forecasting chain; Liechti et al. (2013) and Liechti and Zappa (2016) explored the impact on hydrological prediction of different techniques of

rainfall forecast based on both NWP and radar data; Kyznarova et al., (2012, 2013) tried to use the INCA system (Haiden et al., 2011) precipitation products as input to an hydrological model, evaluating the benefit with respect to extrapolation techniques.

Within this framework, the present study attempts to use in a synergistic way a nowcasting model (PhaSt), a high-resolution NWP model (MOLOCH) and rainfall observations. Rainfall estimates derived from both radar and rainauges

are used for frequent (60 minutes) data assimilation by the NWP model, in order, on the one hand to better reproduce observations in terms of spatial and time location, and on the other hand to improve QPF (Davolio et al., 2017a). A probabilistic nowcasting model is adapted in order to ingest the QPF provided by NWP models. A blending technique based on previous studies (Kilambi and Zawadzki, 2005; Kober et al., 2012; Bowler et al., 2006) is finally applied to smoothly mix nowcasting rainfall scenarios and NWP QPF. The use of forecast rainfall fields in nowcasting can

effectively extend the lead time by several hours, making it suitable for issuing flood and flash flood warnings. The possibility to predict with more accuracy the rainfall fields in input to the hydrological model can improve significantly the accuracy of the hydrological forecast applied on short lead times (Berenguer et al., 2005; Silvestro et al., 2012b). The last module of the present nowcasting chain is represented by the distributed hydrological model Continuum (Silvestro et al., 2013; Silvestro et al., 2015b; Cenci et al., 2016) that is used to transform QPF in streamflow

predictions in a frequent updated Flood Forecasting System (20 minutes). Using as input the forecast rainfall as described above, it provides a probabilistic output, with many predicted discharge scenarios, useful in real time management operations of emergency.

Although the elements of the chain are not innovative by themselves, some new elements are introduced within this study. First, the blending is performed not only combining the rainfall fields forecast by the nowcasting and the NWP

model in their spatial distribution as in more standard approaches (Kilambi and Zawadzki, 2005); in fact the nowcasting rainfall fields are modified along the forecast window according to the information related to the time variation of rainfall volume derived from the NWP model, and this could be consider as a matter of fact, a sort of blending of the rainfall volume. Second, the NWP model forecast is rapidly "updated" with observations trough a data assimilation technique before its information content is used to trigger the nowcasting model. Finally, verification is done in a

hydrological perspective following both a point and a distributed approach in order to enhance the sample of data. The system is applied on three major flood events occurred on the Liguria Region in autumn 2014; results are presented



comparing streamflow forecasts obtained using rainfall predictions from different configurations of the system, in order to evaluate the benefit of using or not some modules.

The manuscript is organized as follows: in Section 2 the area of study and the data used are introduced; in Section 3 the models, the methods and the type of analysis performed are explained. The results of the work are presented in Section 4 while the conclusions and final considerations are drawn in Section 5.

## 2    Study area and input data

The area of study is the Liguria Region; this region, located in north-western Italy (Fig. 1a), is mainly mountainous, it faces the Ligurean Sea and most of its urban areas developed along the coast. The presence of a large number of catchments, characterized by small drainage area (often less than 10 km$^2$) and short response time (few hours at most), with outlet to the Ligurean Sea represents a critical factor that increases the risk of floods.

Therefore, it is clear that high resolution precipitation and hydrological measurements are crucial in urban areas like Genoa and that new approaches to modeling urban catchment properties and hydrological/hydraulic response are needed.

The main input for the hydrological forecasting chain is the observed rainfall. This observed rainfall field comes from the Doppler polarimetric C-band radar, located on Mount Settepani (Fig. 1a) at an altitude of 1386 m, that works operationally with 10 min scansion time and 1x1 km2 spatial resolution. The rainfall field is estimated through the algorithm described in Silvestro et al. (2009), currently used by the Meteorological Weather Services in the Italian regions of Piemonte and Liguria and by the Italian Civil Protection Department. The observed rainfall field is also the input for the nowcasting model and it is used in the assimilation scheme as described in the following Section.

The analysis has been performed according to a distributed approach all over Liguria Region domain but some control sections have been used for a detailed analysis at punctual scale. The relevant sections analyzed are associated to three basins (highlighted in Fig. 1c) mainly affected by the analyzed events of 2014: for 9 October, the Bisagno creek (drained area 97 km$^2$) that flooded the municipality of Genoa during that event; for 11 November, Graveglia (42 km$^2$), a tributary of Entella, has been considered while for 15 November the results regarding Polcevera (140 km$^2$) will be shown.

## 3    Model and algorithms

### 3.1    The meteorological model and the assimilation scheme

The NWP model used in this work is MOLOCH (details in Malguzzi et al., 2006; Buzzi et al., 2014; Davolio et al., 2017b), that integrates the non-hydrostatic, fully compressible equations for the atmosphere, on a latitude-longitude rotated Arakawa C-grid, with a resolution of 0.02 degrees, equivalent to about 2.2 km, and on 60 vertical levels (hybrid terrain-following coordinates). The integration domain (Fig. 1b) covers north and central Italy and initial and boundary conditions are provided at 1-hour interval by the BOLAM model forecasts, as in the current operational practice at CNR ISAC. BOLAM is a limited-area hydrostatic model (Buzzi et al., 2004) based on primitive equations with a convective




parameterization derived from Kain (2004). Initial and boundary conditions for BOLAM (domain shown in Fig.1b) are defined with ECMWF analyses. MOLOCH is nested in a 3-hour forecast of BOLAM to avoid a direct downscaling form the global analysis to the high-resolution grid and runs for 21 hours. BOLAM and MOLOCH differ mainly in the dynamical core, including the fact that MOLOCH resolves explicitly deep convection, while the following parameterization schemes are common in the two models: atmospheric radiation, atmospheric boundary layer and
surface layer, soil processes and, to a large extent, microphysical processes.

The assimilation method, whose implementation in MOLOCH is shown in Fig. 2, is based on nudging. During the assimilation window, model specific humidity profiles at each grid point are progressively modified depending on the comparison between observed and forecast rainfall. To attain this aim, hourly precipitation estimates provided by Settepani radar are used as observations. The assimilation scheme is explained in detailed in Davolio et al. (2017a). The
set up conceived for the present application takes into account the time requirements for an operational implementation. Considering the timing for global data availability and radar estimates delivery and processing, the first assimilation windows covers the first 6 hours of forecast. At the end of this period, the model state is stored and a free forecast is run to cover the following hours. In this way, once a new hourly rainfall observation is available, an additional 1-h assimilation is performed, re-starting the model from the stored condition. This procedure can proceed for several hours,
at least until the following global analysis is available, and allows for updating and improving forecasts, as a consequence of a longer assimilation period.

### 3.2 The nowcasting technique

The nowcasting technique used in this paper is a radar-based probabilistic technique, PhaSt (Metta et al., 2009). PhaSt is a "phase stochastic" spectral-based nowcasting procedure that, starting from two following rainfall fields observed by
radar, realizes a non linear-empirical transformation, and stochastically evolves the fields within the spectral space. The use of the spectral space allows to preserve the spatial correlation within the rainfall fields. The evolution of Fourier phases trough the stochastic process generates many realizations, to be used as members of an ensemble of precipitation nowcasts. All the ensemble members are characterized by the same amplitude distribution and very similar power spectra. However, the phase evolution (i.e., the positioning of rainfall structures) evolves differently in the different
realizations, providing an estimate of the probability of occurrence of precipitation at a given point in space and a given instant in time. The main equations of the algorithm are reported below (Eq. 1):

$$
\begin{cases}
k_s = \left(k_x^2 + k_y^2\right)^{\frac{1}{pwr}} \\
d\Phi_{k_s} = \omega_{k_s} dt \\
d\omega_{k_s} = -\left(\omega_{k_s} - \omega'\right)\dfrac{dt}{T} + \sqrt{\dfrac{2\sigma^2}{T}}\sqrt{1 - \dfrac{dt}{2T}}\, k_s dW \quad (Ornstein - Ulhenbeck\ process)
\end{cases}
\tag{1}
$$

Where:

•        $k_s$ is the spectral phase, dependent on the wavenumbers $k_x$ and $k_y$: through this relation is possible to give more weight to the smaller scales and less weight to the larger ones;

•        T is the decorrelation-time, after which the rainfall field is assumed to stop





- $\sigma^2$ is the variance of the process;

- $dW$ is a random increment drawn from a normal distribution with zero mean and second-order moment (W is a Wiener process).

To allow for the presence of time correlations in the angular frequencies a Langevin-type model is used: the temporal evolution of the Fourier phase $\phi(k_x, k_y)$ at a given wavenumber $(k_x, k_y)$ is written in terms of a linear Ornstein–Uhlenbeck stochastic process for the angular frequency. The Ornstein–Uhlenbeck process generates angular frequencies that have a Gaussian distribution with zero mean and variance $\sigma^2$ and an exponentially decaying temporal autocorrelation. The representation of the noise of the forecast rainfall field is disentangled into two components, one related to a noise constant in time and the other to a noise changing in time and space in order to have different forecast 180 rainfall fields, but coherent with the initial observed rainfall.

### 3.3 Modification of nowcasting technique with NWP information

To improve the nowcasting algorithm, its original formulation (Metta et al., 2009) is modified in a constraint regarding the spectral amplitude of the rainfall field. This spectral amplitude was previously kept constant along the forecast horizon, that means fixing, in the real space, the total volume of the forecast rainfall fields equal to the one of the last 185 observed radar image. This hypothesis of constant volume is relaxed. Therefore, the volume is modified according with the information provided by the NWP model corrected through the data assimilation technique. The volume trend is calculated for the hours of forecast using the information about the total volume of precipitation on the radar domain (Fig. 3) according to the following formula (Eq. 2):

$$Rainfall\ field_{Vol\ Mod}(x, y, en, T) = Rainfall\ field(x, y, en, T)_{PhaSt\ Forec} * Volume \tag{2}$$

Where $x$ and $y$ are the coordinates in the radar rainfall field, *en* identifies the ensemble member (20 members in this 190 application) and $T$ is the lead time of the forecast (from 10 minutes to 6 hours).

In this way the information about the potential growth and decay of the rainfall structures is provided as an additional information to the nowcasting rainfall field. The idea is to make the forecast able to include in a simple way the tendency of decreasing or increasing of precipitation volume, leaving to the original scheme the task to describe spatial and temporal distribution of the rainfall field.

### 3.4 Blending method

The nowcasting models have shown a predictability limit after some hours of forecast. This limit is valid also for the nowcasting model Phast according to previous studies (Metta et al., 2009). This behavior is mainly due to the fact that in the nowcasting model the evolution of the precipitation systems is not included: the physics that represents growth and decay of the precipitation systems becomes progressively more important with increasing lead time (Germann et 200 al., 2006). These physical processes are represented in the NWP model so that it is worth to connect the nowcasting and the meteorological model in a resulting rainfall forecast possibly more accurate. The blending technique tries to overcome this limit, connecting these two components. Blending has been previously analyzed in some studies with the purpose of improving the rainfall forecast (Golding, 1998; Kober et al.,2012; Atencia et al, 2010; Kilambi and





Zawadzky, 2005; Nerini et al., 2018). In this application, a blending function is written in order to balance the forecast reliability of the two models.

Many of the previous methodologies to estimate the blending function started from the statistical indexes computed on the forecast rainfall field for the two models. These indexes allow to calculate the weight for the two different forecast rainfall fields. This strategy cannot be applied in this case due to the scarcity of rainfall events considered: having only three case studies it is not possible to have representative scores for the two methodologies. In this work, being aware of the blending functions already presented in literature and according to the arguments presented before concerning the capacity of the model to represent the physical processes after the first hours of forecast, a first guess of the blending functions is selected as shown in Fig. 4.

It is worth noting that the weight of the NWP model forecast rainfall is calculated as the complementary function (Eq. 3):

$$Weight_{NWP} = 1 - Weight_{NOWC} \qquad (3)$$

The first function, that will be hereinafter referred to as "step function", gives all the weight to the nowcasting rainfall field for the first two hours and then, up to the end of the forecast, to the NWP model, as a switch on/off between the two models. The other three functions are a progressive smoothing of this step function, trying to produce an intermediate forecast rainfall field between the nowcasting and the NWP through a linear combination of them.

At a first glance it could seem that all blending functions give too much weight in the first time steps to the nowcasting model. However, it must be kept in mind that the nowcasting rainfall field is obtained exploiting the information from the NWP model corrected with data assimilation to modify the radar rainfall volume (as described in Section 3.3). Moreover we expect that forecast derived by extrapolation from observation can capture the spatial-temporal pattern better than NWP during the first two hours of forecast (Metta et al., 2009; Collier, 1981; Seed, 2003; Xu and Chandrasekar, 2005; Berenguer et al., 2011). In this way more weight is given to the information of the nowcasting model regarding the positioning of the rainfall structures and a first correction of the forecast is done through the modification of the volume. Then, for longer forecast time, less weight is given to the nowcasting rainfall field in favor of the NWP model forecast. The latter gains progressively more importance up to almost 5 hours ahead. After, the rainfall field is entirely provided by the meteorological model. Hence, the blended rainfall field at a certain forecast time T results from the linear combination as follows (Eq. 4):

$$Rainfall\ field_{blended}(T) = (weight_{NOWC}(T) * rain_{NOWC}(T)) + (weight_{NWP}(T) * rain_{NWP}(T)) \qquad (4)$$

and it is used as input for the hydrological model for six hours of forecast.

Since the forecast of the NWP model is deterministic ($rain_{NWP}$), the rainfall field that is combined with the 20 members generated with PhaSt ($rain_{NOWC}$) is always the same. As a consequence, for short forecast time (values of $weight_{nowc}$ close to 1) PhaSt leads to different rainfall scenarios, while, as long as the forecast time increases (values of $weight_{nowc}$ close to 0), each rainfall scenario will tend to be equal to the only one NWP QPF.



### 3.5 The hydrological model

*Continuum* is a continuous distributed hydrological model that relies on a morphological approach that identifies the drainage network components from a Digital Elevation Model (DEM) (Giannoni et al., 2000; Giannoni et al., 2005). The DEM resolution generally coincides with the model spatial resolution. Flow in the soil is divided in a sub-surface flow component that is based on a Horton schematization (Gabellani et al., 2008) and that follows the drainage network directions, and a deep flow component that moves based on the hydraulic head obtained by the watertable modeling. The energy balance is solved explicitly at cell scale by using the force-restore equation. A complete description of the model is reported in Silvestro et al. (2013).

The hydrological model is implemented at a spatial resolution of 0.005 deg (about 480 m) and with a time resolution of 10 minutes. The implementation and calibration of the model in the testing area is described in details in Davolio et al. (2017a) and Silvestro et al. (2018); the latter works evidenced good values of the employed statistics during the calibration and validation process.

It is worth noting for the scope of the present study that the model is distributed so the streamflow is available in each point of the modeled drainage network; this characteristic is exploited for the verification (presented in the Section 4). Due to the limited horizontal resolution of the model in this application, basins with drainage area lower than 15 km$^2$ are not considered (Silvestro et al., 2018).

### 3.6 The hydrological nowcasting chain

Within the hydrological nowcasting chain all the elements presented in the previous sections are connected together. The main input is the observed radar rainfall, that is used up to the "now" as it is; then it is used to evolve the last radar image within the nowcasting phase and it is used in the data assimilation process of the NWP model. Then, for the rainfall forecasting time window three configuration are considered and compared. These configurations differ for the 6-hour forecast rainfall field that is provided in input to the same hydrological model (Figure 5), which is computed as follows:

1) nowcasting without volume variation and no blending with NWP model (Figure 5a);
2) nowcasting with volume variation according to NWP model with data assimilation; no blending with NWP model (Figure 5b);
3) blending between rainfall fields from nowcasting with volume variation (as in 2) and from NWP model (deterministic forecast) using a blending function varying in time as presented in Section 3.4 (Figure 5c);

In the first configuration, hereinafter referred as to NOWC, the forecast rainfall field is provided for 6 hours solely by the nowcasting system PhaSt (as described in Section 3.2). In the second, that will be referred to as NOWC VOL, the volume variation from NWP is used during the forecast time to modify the rainfall fields (as described in Section 3.3).

Finally, the last configuration (referred to as NOWC BLEND) represents the complete forecasting chain: it uses the observed rainfall fields to build the rainfall scenarios in the recent past and the nowcasting model with volume correction for the first two hours of forecast; then, from the second to the sixth hour, the rainfall field is a linear combination between nowcasting and NWP model outputs, according to the blending function described in Section 3.4.





The NWP is constantly corrected with hourly observation in order to provide updated precipitation volume trend to the nowcasting algorithm.

For all the configurations the hydrological model takes in input the rainfall scenarios and produces the forecast in terms of streamflow. The output of the chain is an ensemble of possible discharge scenarios (20 ensemble) for the following 12 hours.

**3.7  Methodology of analysis**

The comparison against observations is done in terms of the final output of the chain, namely the discharge forecast. This comparison is performed taking as reference the discharge forecast obtained using the radar rainfall estimates as input to the distributed hydrological model (hereinafter referred to as "reference hydrograph"). This approach does not consider errors in the hydrological model, since it is aimed at evaluating possible improvements in rainfall forecast
(Borga, 2002; Vieux and Bedient, 2004, Berenguer et al., 2005).

Before comparing the three configuration described in Section 3.6, a first preliminary analysis is performed to evaluate different blending functions (as described in Section 3.4). This allows to identify the best function for every event and an overall (on average) best function. Then the comparison is done between the three configurations in order to investigate the importance of each element used to build the rainfall scenarios in the streamflow prediction.

A first analysis evaluates the performances of the hydrological forecast in three control sections, one for each considered event. In this analysis three scores are used: the Nash Suttcliffe coefficient (Nash and Sutcliffe, 1970), the Variance of the discharge (Var) and the Continuous Rank Probability Score.

The Nash Sutcliffe (NS, see Eq. 5) coefficient is chosen since it is one of the widely used measures to evaluate model performances, especially for streamflow reproduction.

$$NS = 1 - \frac{\sum_{t=1}^{T}(Q_m(t) - Q_{obs}(t))^2}{\sum_{t=1}^{T}(Q_{obs} - \overline{Q_{obs}})^2} \qquad (5)$$

Where $Q_m(t)$ e $Q_{obs}(t)$ are the modeled and observed streamflow at time t. Using a probabilistic forecast the index is calculated for each of the 20 realizations and then a mean value is taken.

To relate this index to the spread of the ensemble of discharge forecast, the variance is calculated for the analysis on punctual section (Eq. 6).

$$Var(X) = E[(X - \mu)^2] \qquad (6)$$

Where X is the forecasted discharge and μ is the mean of the forecast.

The Reduced Continuous Rank Probability Score (RCRPS, Eq. 7) (Trinh et al., 2013) is computed as the CRPS (Brown, 1974; Matheson and Winkler, 1976; Unger, 1985; Stanki et al., 1989; Hersbach, H., 2000), reduced with the standard deviation of the observed discharge over the analyzed time period (hereafter $\sigma^2$).




$$RCRPS(F, x) = \frac{1}{\sigma^2} \int_{-\infty}^{\infty} (F(y) - \mathbb{1}(y - x))^2 dy \qquad (7)$$

Where $F(y)$ is the forecast probability CDF for the forecast and $\mathbb{1}$ is the step function of the observed value. A value of RCRPS equals to zero means a perfect forecast: observations and forecast coincide. Increasing values corresponds to a

bigger distance between observations and forecast.

NS and Var(X) are applied on the mean of the streamflow ensemble following a deterministic approach in the comparison, while the RCRPS is used to evaluate results in a probabilistic perspective (Trinh et al., 2013; Davolio et al., 2017a). The values of all the scores are expressed as function of lead time; to cope with the large number of values of the RCRPS its visualization has been done using boxplot, as it will be shown in Section 4.

The distributed analysis is carried on exploiting the distributed maps of discharge produced by Continuum. This allows to use several sampling points for the comparison even if only three events are analyzed. In this case a first distinction is done dividing all the interested points according to their associated drained area into three classes:

- Points with upstream drainage area in the range of 15 to 50 km$^2$ (small catchments)
- Points with upstream drainage area in the range of 50 to 150 km$^2$ (medium size catchments)
- Points with upstream drainage area in the range of 150 to 500 km$^2$ (big catchments)

This distinction finds a motivation in the concentration time related to the different size of catchments: a longer-lasting influence of the forecast rain can be found for the basins with greater drained area and then bigger concentration time. Hence, analyzing the performance indexes and relating them to the lead time, a longer lead time will be considered in calculating the indexes for the bigger catchments. In particular beyond the 6 hours of lead time corresponding with the

rainfall forecast a further window of forecast discharge is considered: 1 additional hour for the 1[st] class of catchments, 2 additional hours for the 2[nd] and 3 additional hours for the 3[rd].

The distributed analysis, executed on the entire domain on which the hydrological model is run, provides more general information about the performance of the hydrological nowcasting chain. The score used for the distributed analysis is the RCRPS, calculated for each point of the domain interested by the event (a filter on the pixel in which the forecast

discharge does not exceed a threshold is performed). Also the RCRPS is expressed as a function of lead time and for the three classes of drained area.

## 4    Results

### 4.1    Impact of the blending function on hydrological forecasts

A first analysis is performed to evaluate the performance of the complete chain implemented using four different

blending functions to linearly combine nowcasting and NWP rainfall fields. This analysis is done evaluating the discharge forecast using the RCRPS score; the distributed approach is applied, hence the predictions in all the points of the domain are considered. The results are then analyzed for the three classes of drainage area presented in Section 3.7.

Figure 6 presents a summary of the results concerning the entire sample of data for the three events: in this boxplot each group of column represents the values of RCRPS (y axis) for the indicated lead time every 20 minutes (x axis). Each





box indicates the values of the 25% and 75% quantiles, the horizontal line inside the box represents the median value, while the circle indicates the mean. The whiskers extend to the most extreme data points not considered outliers, and the outliers are plotted individually with points. The four columns represent the results related to the use of the different blending functions: the first column (red) is related to the step blending function; the second (green, hereafter defined as f1) and the third (yellow, f2) refer to the use of intermediate functions; the last column (blue, celled f3) is related to the

smoothest blending functions, as presented in Section 3.4. Results show clearly that there are no large differences among the four configurations of the blending functions, but in general it seems opportune to rapidly move from nowcasting to NWP model forecast avoiding long smoothing periods. In fact the scores for all the classes of drainage area indicate a worse performance (large values of RCRPS) around 4-5 hours of lead time. This result is probably due to the fact that, even if data assimilation in the NWP model is performed with hourly updating, often Phast rainfall

scenarios and NWP rainfall fields are not seamless. These discontinuities can affect the blending process, generating an unrealistic final rainfall scenario. In general the best score is obtained with the step function. However, this is an averaged behavior for the three events; each event has its own best blending function, as will be shown in Section 4.2 and 4.3.

### 4.2  Basin scale analysis at river sections

The results regarding the discharge forecasts for the main basins stroke by the analyzed events are examined starting from a first qualitative visualization of the comparison between the hydrographs for each configuration at every time step (every 20 minutes). Two examples are shown in Figure 7. The first one is the forecast at 20:10 UTC on 9 October 2014 on Bisagno creek, that was the main responsible for the flood of the municipality of Genoa during this event. The second one is the forecast at 19:40 UTC on 11 November 2014 for Entella basin, mainly interested during this event

with its tributaries Graveglia and Lavagna. The figures show the envelopes of the discharge forecasts for one-time step during the considered time window of the event. The black thick line is the reference hydrograph, while the dashed lines represent the mean of the discharge ensemble forecast for each lead time. The first hydrograph envelope (light blue) refers to the configuration NOWC obtained with 6 hours of nowcasting without volume modification. For the second hydrograph (orange), the 6 hours of rainfall are those predicted by the configuration NOWC VOL. For the last

hydrograph (red) the input rainfall field results from the blending (configuration NOWC BLEND), using the mean best function for the three events identified in Section 4.1, that is the blending step function.

Results clearly indicate that the spread of the discharge forecast ensemble is markedly smaller when input rainfall is provided by blending instead of nowcasting alone (with or without volume trend modification). This is possibly ascribable to the fact that while nowcasting provides a probabilistic forecast (20 ensemble) of different rainfall

scenarios, blending connects a deterministic forecast from the NWP model with nowcasting ensemble. Since each member of this ensemble is blended with the same NWP model forecast the spread of the final ensemble is smaller.

Figure 8 summarizes the results for some representative basins for each event, using the statistical scores described in Section 3.7. The three configurations of the hydrological chain presented in Section 3.7 are compared; for the blending configuration, the best blending function (the blending step, see section 4.1) and the local best blending function, which

is different for each event, are applied.





The event of 9 October is sadly known for the flood that affected the municipality of Genoa during the evening. The main creek interested by the flood was Bisagno and the results at the Passerella Firpo section (A≈97 km$^2$) are shown in Figure 8 (top row). While for the NS the performances of the hydrological nowcasting chain are really similar among each other, the variance of the forecast obtained using the blending is smaller than that obtained using nowcasting, thus providing a forecast with less variability, which could be an advantage or a disadvantage depending on the single case. The RCRPS score shows a variable behavior for different lead times: at the beginning, the best performing configuration is NOWC VOL; in the intermediate phase, when nowcasting and NWP model forecast are combined with blending, there is no a configuration overperforming the others; for longer lead times, the configuration NOWC produces the best performance. For this event, the forecast of the meteorological model, even corrected with data assimilation, is not able to improve the QPF.

Although the event of 11 November affected the whole Liguria Region, the main effects at the ground were caused by the Entella river, that flooded the urban area of Chiavari, and by its tributaries Lavagna and Graveglia. The scores on Graveglia basin at Caminata section (A≈42 km$^2$) are shown in Figure 8 (middle row). Conversely to the previous case, here the use of the NWP model has a clearly positive impact. In this event the rainfall fields from nowcasting techniques lead to an overestimation of the discharge, while the rainfall fields obtained through the blending improves markedly the discharge forecast.

For the event of 15 November, the performance of the system is evaluated for Polcevera at Rivarolo section (140 km$^2$) and shown in Figure 8 (bottom row). It is worth noting that the RCRPS boxplot shows that the performance of the system fed with blending becomes worse between 4 to 6 hours lead time. As already explained this can be related to the unrealistic rainfall field produced by the blending when the nowcasting and the NWP model forecast differ too much between each other.

### 4.3 Distributed analysis

While in the previous Section the scores were computed for single punctual sections of the basins, here the distributed analysis is carried out, aimed at giving a more general picture of the performance of the hydrological nowcasting chain fed with different rainfall inputs. In fact, with the distributed analysis it is possible to compute the score, in this case the RCRPS, over all the points of the domain, increasing the data sample used for the analysis.

**9 October 2014 event**

For this event the best blending function (Section 3.4) is the f3. Figure 9 shows the general behavior during the entire event: the use of the information retrieved by the NWP model in the rain forecast worsens the hydrological forecast. This can be due to the peculiar type of event characterized by stationary and persistent heavy precipitation on the same portion of territory that was not forecast precisely by the NWP model, but well reproduced by the nowcasting model. However, even if the information regarding the location of the rainfall, coming from the meteorological model, is misleading for the hydrological forecast, the information about the total volume on the domain is adding a value to the nowcasting rainfall field.

**11 November 2014 event**





For this event the best blending function is the f2. As already pointed out in the previous basin scale analysis, the distributed analysis confirms, even more clearly, that for this event the system using the blending performs markedly better. Especially for the bigger basins, due to their response time, the effects of a proper rainfall forecast provided with blending are beneficial for longer lead times, probably due to the slow response of the basins.

**15 November 2014**

For this event the best blending function coincides with the best in average for all the three events, that is the step function. In this case the differences among the configurations can be noticed most of all in the first two classes of area, where the use of the blending technique improves the rainfall forecast: the column related to the NOWC BLEND configuration is always lower and presents less spread with respect to NOWC and NOWC VOL configurations. In the last class of area (larger basin) the behavior is different especially at the lead time corresponding to the transition in the blending between the rainfall field from nowcasting and from the NWP model. This transition is confirmed to be the most critical phase for the blending as it can produce unrealistic rainfall fields.

## 5    Discussion and conclusions

A hydrological nowcasting chain is a useful instrument for flood and flash flood warnings. The use of an accurate QPF in input to the hydrological model is essential to extend the lead time of the hydrological forecast. The aim of this study is to improve the single elements in order to provide in real time an accurate forecast of the rainfall field, able to improve the performance of the hydrological prediction at a temporal scale up to 6-8 hours.

The elements involved in the chain are the high-resolution NWP model MOLOCH, the nowcasting model PhaSt and the hydrological distributed model Continuum. To improve the NWP forecasts, the model is frequently (every hour) corrected with data assimilation of rainfall estimates derived from both radar and raingauges. Then the forecast rainfall fields produced by PhaSt are modified along the forecast horizon according to the information on the variation of rainfall volume derived from the NWP model corrected with assimilation. This is a first attempt of blending within the nowcasting, aimed at taking into account the processes of growth and decay of the precipitation structures, that progressively gain importance at increasing lead time. Then a standard blending technique is applied to linearly combine the rainfall fields from nowcasting and NWP model, based on a blending function that is giving different weight to the QPF depending on the lead time. The probabilistic QPF obtained (20 rainfall ensembles) is the input of the distributed hydrological model Continuum that produce an ensemble of discharge forecasts in a frequent updated Flood Forecasting System (every 20 minutes).

The meteo-hydrological chain is tested for three main floods occurred during the autumn of 2014 that affected different areas of Liguria Region: 9 October event with the flood of Bisagno creek that hit the municipality of Genova; 11 November event, involving Graveglia catchment; 15 November event, in which the flood of Polcevera occurred. Even if the number of analysed events is restricted, the resulting distributed maps produced by Continuum allow to verify the performance of the chain on a large data sample.

A first analysis has compared the results of the application of various blending functions to combine the forecast fields, highlighting the presence of a best function on average for the three events and a best function for each single event.





Then a comparison between three configurations of the hydrological nowcasting chains is performed. In the first two configurations, rainfall input is provided by nowcasting, with or without volume modification, while in the third one the blended rainfall fields are used.

Statistical scores show that in various cases the use of the rainfall fields resulting from the blending process leads to an improvement of the performances of the whole chain with respect to the use on the nowcasting alone. In other cases the benefit gained using the complete configuration with the blending is not so evident and the performances result similar to the use of the nowcasting for all the lead time. However, a worsening of the performance is rarely observed and occurs in the time window corresponding to the transition between nowcasting and NWP model forecast rainfall field. Hence, there is an added value in the use of the blending between nowcasting and NWP model as it produces better or

equal scores with respect to the use of the nowcasting alone.

Future works and improvement to the chain presented in this work will be explored always in terms of improvement of the elements that are composing it. Other technique of data assimilation with increasing degree of complexity can be used to assimilate in the NWP model not only the observed precipitation field but also other variables. Following recent works (Atencia et al., 2010) further investigations can be done using another type of blending, called spatial blending,

that introduces spatial dependence of weights as distance function to rainfall structures.



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





**Figures**

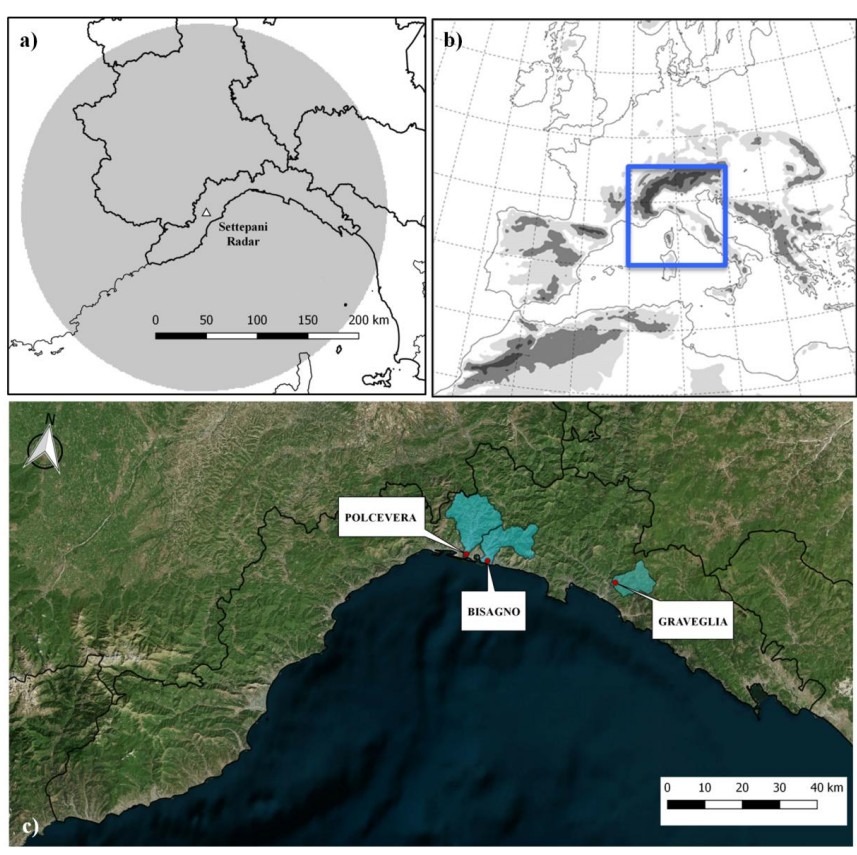

**Figure 1:** Area of interest: a) Location of the Radar in Liguria Region and area covered; b) BOLAM and MOLOCH (blue square) integration domains; c) Liguria Region and the drained area of the basins analyzed for the three events: Polcevera and Bisagno, flowing inside the urban area of Genoa and Graveglia, one of the main tributaries of Entella basin, the biggest basin of Liguria Region.







**Figure 2:** MOLOCH forecasts and data assimilation implementation. The first assimilation is performed during the first 6 hours of forecast, while after that a 1-hour assimilation is done restarting the model from the stored conditions of the previous run (dots).

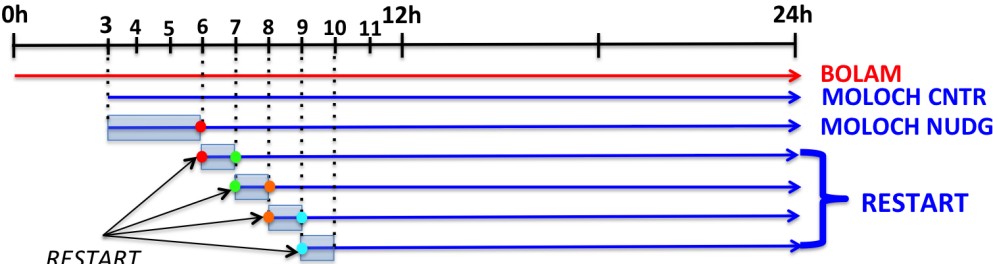


**Figure 3:** Volume trend for rainfall field modification (example for 10 Nov. 2014 at 12:00 UTC). The total volume on the domain considered is summed for each time step of MOLOCH forecast with DA. The trend volume is applied to the first rainfall forecast by the nowcasting technique.




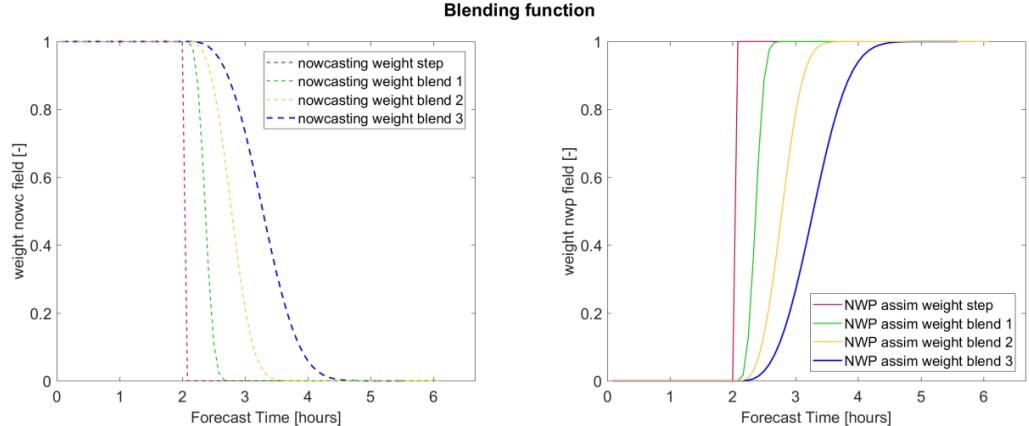


**Figure 4:** Different blending functions analyzed: on the left the weighting function applied to the nowcasting rainfall field; on the right the complementary function, used to weight the NWP model forecast. The first weighting function (red line) is a step function while the other functions are increasingly smoother.

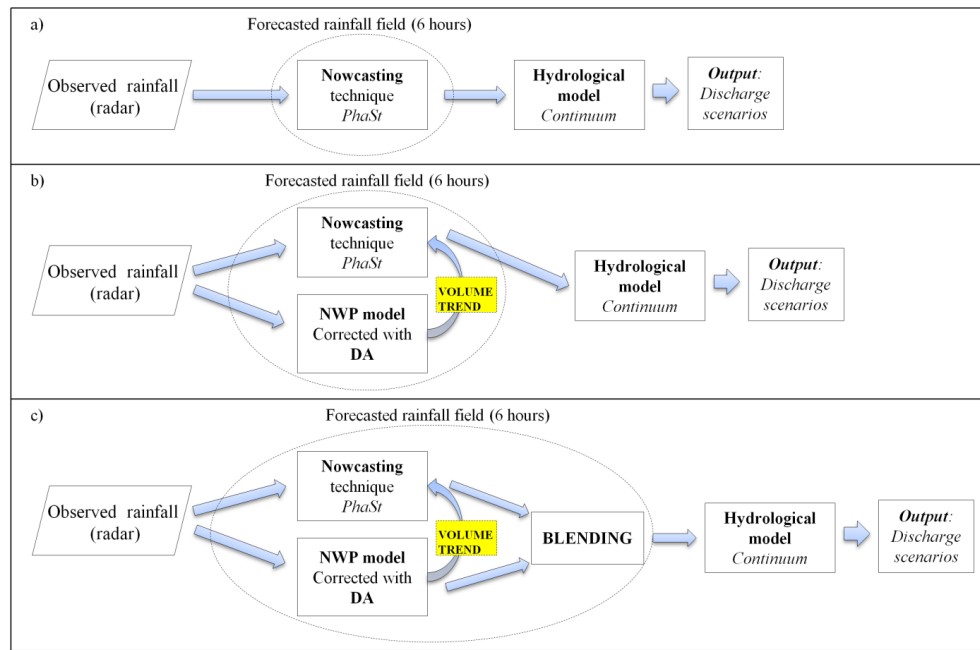


**Figure 5:** The hydrological nowcasting chain in its three configurations: the first one in which is used only the nowcasted rainfall field without volume modification (a); the second one with the nowcasted rainfall field modified with the trend retrieved by the NWP model (b); in the last one the nowcasted rainfall fields with volume modification are combined through blending with the fields forecasted by NWP model corrected with DA





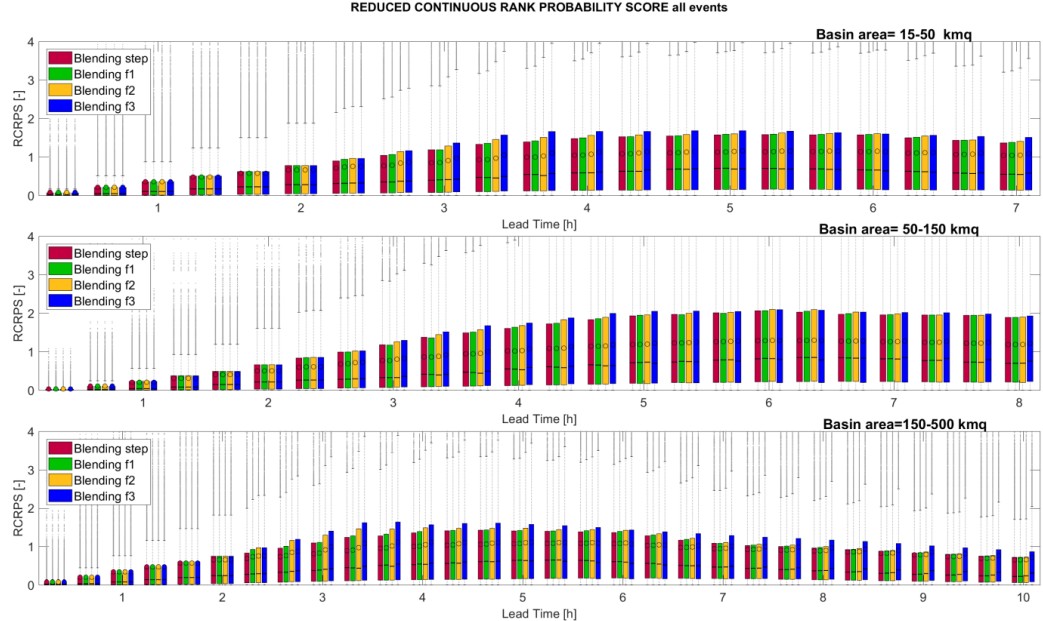


**Figure 6:** RCRPS for all the events: analysis of the discharge output with the four different blending functions. The red column represents the results obtained by applying the step function, the other three columns refer to the other three smoothed blending functions (see Section 3.4). Inside each box, the circle indicates the mean value, while the line indicates the median. The results are presented in form of boxplot for the three classes of drainage area (15-20 km$^2$, 50-150 km$^2$ and 150-500 km$^2$).








**Figure 7:** Example hydrographs for 9 October and 11 November event, for Bisagno and Entella outlet sections, respectively. In both figures. discharges are obtained using as input the 6 hours rainfall field from NOWC (light blue), NOWC VOL (orange) and NOWC BLEND (red).





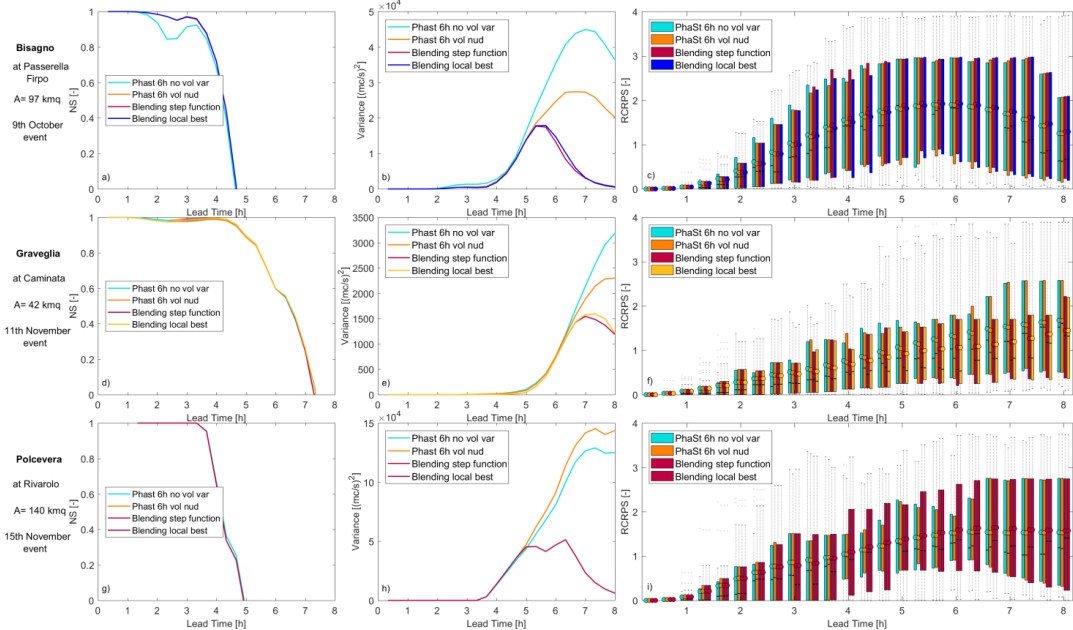

**Figure 8:** Analysis at basin scale for the three events, namely 9 Oct. 2014 on Bisagno creek (top), 11 Nov. 2014 on Graveglia (middle) and 15 Nov. 2014 on Polcevera (bottom). The following scores are shown: Nash Sutcliffe (NS) efficiency (left column), Variance (middle column) and RCRPS (right column). Four configurations of the system are evaluated (see text).

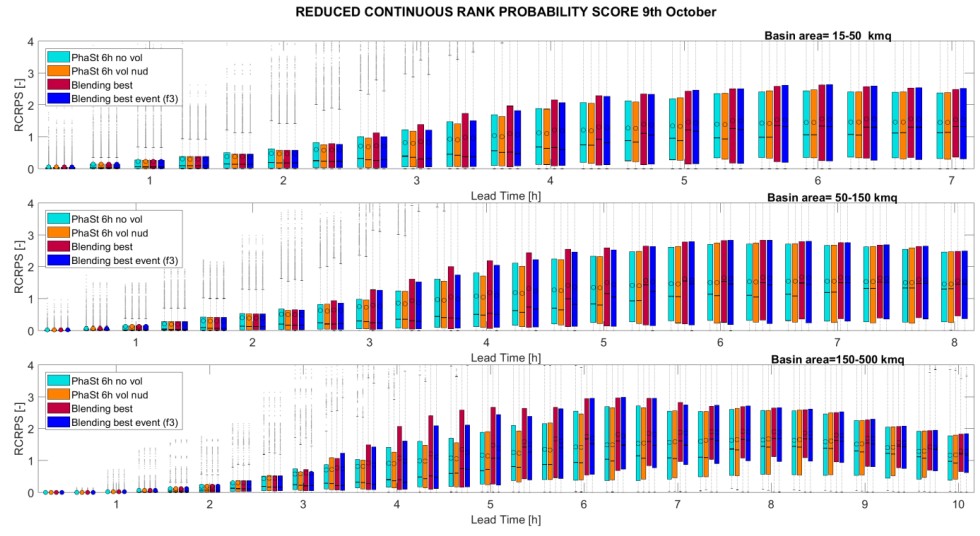

**Figure 9:** 9 Oct. 2014 event: RCRPS for three distinct classes of area. Each column refers to a different configuration of the forecasting system: NOWC (light blue), NOWC VOL (orange), NOWC BLEND using the step blending function (red), and NOWC BLEND using the best local blending function (blue) which in this case is the function f3.




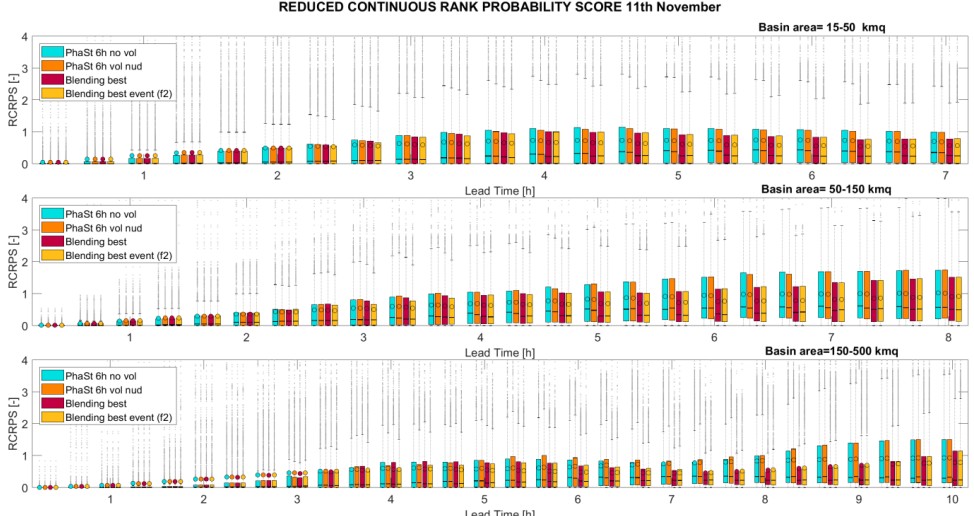

**Figure 10:** 11 Nov. 2014 event: RCRPS for three distinct classes of area. Each column refers to a different configuration of the forecasting system: NOWC (light blue), NOWC VOL (orange), NOWC BLEND using the step blending function (red), and NOWC BLEND using the best local blending function (yellow) which in this case is the function f2


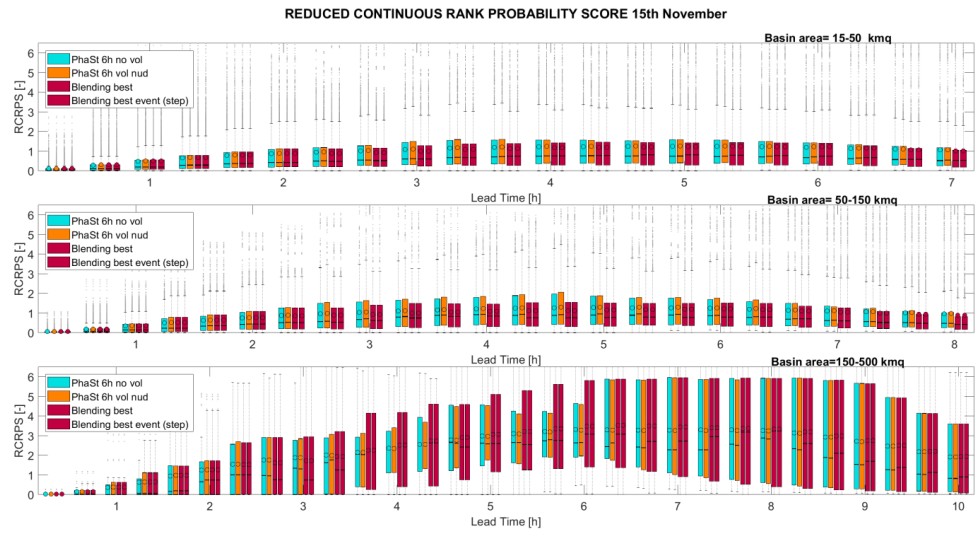

**Figure 11:** 15 Nov. 2014 event: RCRPS for three distinct classes of area. Each column refers to a different configuration of the forecasting system NOWC (light blue), NOWC VOL (orange), NOWC BLEND using the step blending function (red), and NOWC BLEND using the best local blending function which in this case coincide with the step function (red).
