# Peer review of "Using nowcasting technique and data assimilation meteorological model to improve very short range hydrological forecasts"

_Hydrology and Earth System Sciences, 2019_

## Referee Comment (RC1) · Anonymous Referee #1 · 1 Apr 2019

**General comments**

The paper entitled "Using nowcasting technique and data assimilation in a meteorological model to improve very short range hydrological forecasts" by Poletti et al., aims at exploiting both observations and modelling sources to improve the discharge prediction in small catchments with time horizon of 2-8 hours. In particular, observations are used in a frequently updated data assimilation framework to drive the NWP system, whose output is in turn used to improve the information in input to a nowcasting technique; finally, nowcasting and NWP outputs are blended, generating an ensemble of rainfall scenarios used to feed the hydrological 20model and produce a prediction in terms of streamflow.

The paper shows a good literal review, and it is quite well written and readable. However, I strongly suggest to check the punctuation marks which could be much improved.

The research is innovative and it fills into the aims of the paper. I found very interesting how the authors try to merge all observations, radar and NWP forecasts with the blending technique to obtain the best hydro-meteorological simulations. This could be encouraging in the scientific research, above all for flash floods forecast in small catchments as they are in the Liguria Region with a fast response to the rainfall input.

The abstract and conclusions are satisfactory, with scientific methods and assumptions valid and clearly outlined, but I recommend to take care about figures, since some improvements can be done, as above suggested.

**Specific comments**

P6, L189: Please, can you clarify better the ensemble members introduced in this part of the text?
P7, L224-229: Less or more weights according to the forecast hours is a progressive gain as written in the text. Are they related to the different blending functions in Figure 4?
Figure 1: Please, highlight the Liguria border with a different colour.
Figure 2: Please, add "top" and "bottom" in the caption for the figure description.
Figures 6, 9, 10, 11: What is shown over the upper whisker of the coloured box plots?
Please, try to use a greater font for x/y-axis label and legends in the figures.

**Technical corrections**

P2, L34: Add a comma, after (2016)
P3, L71: I suggest: "…and, hence, reduce…"
P3, L79: Add a comma, after (2013)
P3, L84-85: I suggest: "in order to better reproduce observations in terms of spatial and time location, on the one hand, and to improve QPF (Davolio et al., 2017a) on the other hand."
P3, L89: Add "the" before NWP QPF.
P3, L99-102. Please clarify better this sentence. I suggest: "First, the blending is performed not only combining the rainfall fields forecast by the nowcasting and the NWP model in their spatial distribution as in more standard approaches (Kilambi and Zawadzki, 2005), but the nowcasting rainfall fields are modified along the forecast window according to the information related to the time variation of rainfall volume derived from the NWP model…"
P4, L111: Add a comma before "while"
P4, L113: I suggest: "The area of study is the Liguria Region, located in north-western Italy (Fig.1a) …
P4, L122: "km$^2$"
P4, L126: I suggest: "…all over the Liguria Region domain, but…
P4, L128: I suggest: "of 2014: in particular, …"
P4, L130: Add a comma before "while".
P5, L164: I suggest: "evolves differently in the altered realizations,"
P6, L197: Please, check the acronym "PhaSt" how it written in the whole text. Please homogenize it.

P8, L254: Add a comma before "and".
P8, L255: "configurations"
P11, L341: Add a comma after "In general"
P11, L358: Add a comma before "instead"
P11, L359: Add a comma before "while"
P11, L361: Add a comma after "forecast"
P12, L384: Add a comma after "explained"
P13, L 430: please homogenize the Italian words in English as in the rest of the text.
P13, L 432: Add "the" before Continuum
P14, L440: Add a comma after "cases"
P14, L446: "improvements"
P14, L447: "techniques"
P14, L448: Add a comma before "but"
P14, L449: Add a comma before "further"

Captions of Figure 1: I suggest: "Area of interest: a) Location of the Radar in the Liguria Region and its covered area; b) BOLAM and MOLOCH (blue square) integration domains; c) the Liguria Region and the drained area of the analysed basins for the three events: the Polcevera and Bisagno, flowing inside the urban area of Genoa and the Graveglia, one of the main tributaries of the Entella basin, the biggest basin of Liguria Region"

---

## Referee Comment (RC2) · Massimiliano Zappa (Referee) · 23 May 2019

The scope of the paper is timely relevant and contributes to the topic of flash-flood forecasting in small areas. I particularly like the new features added to the Metta et al. (2009) nowcasting technique. The newly introduced relaxation of the volume constrain is a good idea and worth being evaluated.

Already in the abstract the main issue of this manuscript is raised, namely only three major events in a specific area are evaluated. There is no chance to detect false alarms

of such an approach. We work on similar topics and approaches (e.g. Antonetti et al, 2019) and are always requested to provide justification when we use a limited number of events.

Another issue I want to be addressed is the "distributed analysis", which considers different basin classes but show no distributed results. I would expect a map of the target area which spatial visualization of the index of agreement chosen.

The section discussion and conclusion need a major re-arrangement, as no actual discussion is presented.

Best regards

Massimiliano Zappa

See commented PDF for additional inputs.

References: Antonetti, M., Horat, C., Sideris, I. V., and Zappa, M.: Ensemble flood forecasting considering dominant runoff processes – Part 1: Set-up and application to nested basins (Emme, Switzerland), Nat. Hazards Earth Syst. Sci., 19, 19-40, https://doi.org/10.5194/nhess-19-19-2019, 2019.

Please also note the supplement to this comment:
https://www.hydrol-earth-syst-sci-discuss.net/hess-2019-75/hess-2019-75-RC2-supplement.pdf

**Supplement:**

[revised manuscript text omitted]

---

## Author Comment (AC1) · 21 Jun 2019

Thank you for the comments and the suggestions, in the following we report the answers to the specific comments.

P6, L189: Please, can you clarify better the ensemble members introduced in this part of the text?

Specification added regarding the number of members of the ensemble (20)

[Figure]

P7, L224-229: Less or more weights according to the forecast hours is a progressive gain as written in the text. Are they related to the different blending functions in Figure 4?

Yes, they are. The function defined as step function is the one drawn in red, while the green, yellow and blue line represent those functions related to the smother blending functions.

Figure 1: Please, highlight the Liguria border with a different colour. Figure 2: Please, add "top" and "bottom" in the caption for the figure description.

Figures modified, thank you for the comments.

Figures 6, 9, 10, 11: What is shown over the upper whisker of the coloured box plots? Please, try to use a greater font for x/y-axis label and legends in the figures.

Over the upper whiskers are shown the outliers points (over 75% percentile); labels and legends of the figures 6, 9, 10 and 11 have been modified using a greater font.

All the technical corrections (commas, grammar errors) have been accepted and integrated into the text.

Best regards, Maria Laura Poletti, on behalf of the co-authors

---

## Author Comment (AC2) · 21 Jun 2019

Thank you for the corrections and the valuable comments even along the manuscript. The answers to the specific questions are hereafter reported.

"Already in the abstract the main issue of this manuscript is raised, namely only three major events in a specific area are evaluated. There is no chance to detect false alarms of such an approach. We work on similar topics and approaches (e.g. Antonetti et al,2019) and are always requested to provide justification when we use a limited

number of events. "

We know that the limited number of events is something to work on. We will insert a sentence in the conclusion highlighting the fact that the number of analyzed event is small, and that it is not possible to make a detailed evaluation of the system performances also in terms of false alarms. We could do the analysis on only 3 events because of limits on computational resources and on the availability of data.

"Another issue I want to be addressed is the "distributed analysis", which considers different basin classes but show no distributed results. I would expect a map of the target area which spatial visualization of the index of agreement chosen. "

The distributed results, shown in Figure 9, 10 and 11, are represented in terms of RCRPS using boxplot. Each boxplot is representative of many points, grouped according to the lead time of the forecast for which the score is calculated. The use of this representation instead of a map is aimed to summarize all the information regarding the configurations of hydrological nowcasting chain analyzed, all the forecast time steps, the different events. If the reviewer believes that it could be useful a representation of the RCRPS results over the domain upon which the score is computed, it can be added a map, fixed the configuration analyzed, the event considered and the lead time, containing at each grid point a mean value of the RCRPS calculated on all the forecasts of the event. This is not in our opinion really representative of the results, which are probably better shown by the box plots of RCRPS in Figures 9 to 11; moreover it would be necessary to do maps on different lead times, increasing the number of figures; maybe an alternative could be adding a map that shows the grid points which belong to each drainage area class, in order to better interpret the aforementioned figures (9 to 11).

"The section discussion and conclusion need a major re-arrangement, as no actual discussion is presented"

The section presenting the discussion and the conclusion regarding the results and

possible future works will be deepened and extended.

The technical corrections along the manuscript have been integrated.

Best regards,

Maria Laura Poletti, on behalf of the co-authors

---

## Author Response (AR1)

Dear reviewers and dear Editor,

We uploaded a revised version of the paper. We modified the manuscript according to most of the reviewers requests. Various modifications were done along all the manuscript but we report the main structure changes:

- Two figures have been introduced in the text (now Figure 6 and Figure 8). The first one aims to answer to the comments of the Reviewer 2 and of the Editor: it shows the river network grid points on which the distributed analysis is performed. Each cell is colored according to the class drainage area to which it belongs. Figure 8 is introduced to answer to a specific comment of Reviewer 2 and shows an example of application of the blending technique.
- The section Discussion and conclusion has been re-elaborated according to the comments of Reviewer 2 and of the Editor.
- The main good points and limitations of the study have been pointed out much more clearly both in the abstract and along the text.

Answers to the reviewer comments are reported in the following, together with a description of the modifications to the manuscript. In the manuscript the main modifications are marked in yellow.

For Reviewer 1:

- Figure 1 has been modified according to the Reviewer comment.
- All the figures have been checked and bigger font for label axis and legends have been introduced where needed
- All the technical corrections (commas, grammar errors) have been accepted and integrated into the text.

For Reviewer 2:

- Suggestions along the text have been accepted and integrated in the text, inconsistencies corrected, missed references to suggested interesting paper added;
- The section Discussion and conclusion has been modified according to the comments, trying to extend it and to better highlight the good points and the limitations of the study.
- Comment page 3 line 106: in this study the methodology is applied only to these three events of autumn 2014, but it will be the object of future analysis through the application on other, even more recent, events.
- Comment page 4 line 127: sentence rewritten to clarify the domain of application of the methodology.
- Comment page 5 line 154: the time window of application of the procedure has been clarified in the paragraph.
- Comment page 6 line 186: up to now there is not a deep analysis regarding the point mentioned by the reviewer upon the verification of the technique of modification of the volume; a detailed analysis in terms of rainfall fields may be an interesting point to be developed for further studies, but it's out of the scope of the present paper.
- Comment page 6 line 190: Sentence above the relaxation of the volume added in the abstract as suggested.
- Comment page 8 line 249: sentence rewritten to clarify doubts regarding the analyzed basins.
- Comment page 8 line 263: in paragraph 3.6 the description of the three main configurations has been rewritten to avoid repetitions.

45 - Comment page 9, line 288: we inserted in the text the reference to a previous study upon which the calculation of the NS efficiency as a function of lead time is based (Berenguer et al., 2005).
- Comment page 10 line 305: we added a new figure (now Figure 6) showing the grid points of the river network on which the distributed analysis is performed, distinguishing in terms
50 of class drainage area.
- Comment page 11 line 331: we rewrote the sentence to make the concepts clearer.
- Comment page 11 line 339: a new figure (now Figure 9) is introduced to explain better the sentence, as suggested by the reviewer;
- Comment page 12 line 370: further explanation of the concept.

55 For Editor:

Some main comments (small sample of events considered, rearrangement of the conclusion, new figure reporting the grid points object of the distributed analysis) have been already addressed above.

Furthermore, Figure 7 (now Figure 9) has been modified including the observed discharge, as
60 suggested by the Editor.

Finally, the missing crossed references of Figures 10 and 11 have been added along the text, thanks for having pointed this out.

We hope the paper is now suitable for publication.

65 The authors.

[revised manuscript text omitted]